# Atypical Carcinoid of the Thymus: Early Diagnosis in a Case Report

**DOI:** 10.3390/medsci13030096

**Published:** 2025-07-24

**Authors:** Antonio Mier-Briseño, Miguel Armando Benavides-Huerto, Ismael Padilla-Ponce, Francisco Alejandro Lagunas-Rangel

**Affiliations:** 1Department of Pneumology, Morelia Medical Clinic, Morelia 58260, Michoacán, Mexico; 2Laboratory of Pathology and Cytopathology “Dr. Miguel Benavides”, Morelia 58260, Michoacán, Mexico; 3Department of Surgical Oncology, State Cancer Care Center, Morelia 58020, Michoacán, Mexico; 4Department of Surgical Sciences, Uppsala University, 752 36 Uppsala, Sweden; 5Department of Genetics and Molecular Biology, Centro de Investigación y de Estudios Avanzados del Instituto Politécnico Nacional, Mexico City 07360, Mexico

**Keywords:** thymic neuroendocrine tumor, asymptomatic, thoracic computed tomography

## Abstract

**Background**: Atypical carcinoid of the thymus is an exceptionally rare neuroendocrine tumor originating from neuroendocrine cells within the thymus. These tumors often present with no symptoms or with nonspecific clinical signs, making early diagnosis particularly challenging. Despite their rarity, atypical carcinoids are clinically significant due to their aggressive nature and relatively poor prognosis. Early detection and appropriate management are therefore crucial to improving patient outcomes. **Results**: In this report, we present the case of a 64-year-old patient in whom an atypical carcinoid of the thymus was incidentally discovered following a thoracic computed tomography scan performed for unrelated reasons. Imaging revealed a suspicious anterior mediastinal mass, which was subsequently surgically resected. Histopathological examination, supported by immunohistochemical analysis, confirmed the diagnosis of an atypical carcinoid of the thymus. The tumor demonstrated coexpression of epithelial and neuroendocrine markers, consistent with this rare entity. **Conclusions**: This case adds to the limited body of literature on atypical carcinoid of the thymus and highlights the importance of considering this diagnosis when evaluating anterior mediastinal masses. It also underscores the value of thorough radiological and pathological assessment in identifying early-stage disease, which may significantly influence prognosis and therapeutic strategies.

## 1. Introduction

Neuroendocrine tumors (NETs) are malignant neoplasms that arise from neuroendocrine cells, which are widely distributed throughout various tissues and organs [1]. NETs most commonly arise in the gastrointestinal tract, accounting for approximately 68–74% of cases. The respiratory tract is the second most frequent site, representing about 25%. In contrast, thymic NETs are exceedingly rare, comprising only around 2% [2]. Based on their histological characteristics and degree of malignancy, thymic NETs are classified into four main subtypes: typical carcinoid, atypical carcinoid, large cell neuroendocrine carcinoma, and small cell neuroendocrine carcinoma [3].

Atypical carcinoid of the thymus is a rare neuroendocrine tumor arising from cells of the diffuse neuroendocrine system within the thymus [4]. Its estimated annual incidence is approximately 0.18 cases per 1,000,000 individuals [5]. To date, just over 100 cases have been published worldwide, underscoring its rarity.

Clinical presentation is often nonspecific. Approximately 50% of patients are asymptomatic at the time of diagnosis, and when symptoms do occur, they may be vague, such as cough, chest discomfort, or dyspnea [6]. The tumor’s deep anterior mediastinal location can further contribute to delayed detection, as it may remain clinically silent and radiographically occult in early stages [7]. Despite their indolent presentation, atypical carcinoid of the thymus tends to follow an aggressive clinical course. Local recurrence or distant metastasis has been documented in 20–30% of cases [8]. Reported 5-year overall survival rates range from 56% to 77%, depending on stage at diagnosis and completeness of surgical resection [2].

Given these difficulties, early recognition and diagnosis of atypical carcinoid of the thymus is essential to improve patient outcome. This case contributes to the limited number of documented cases and provides clinical, radiologic, and histopathologic information that may aid in the future recognition and treatment of this rare tumor.

## 2. Case Report

A 64-year-old married male schoolteacher presented to the clinic with a 6-day history of a persistent tickling sensation in the hypopharynx, accompanied by paroxysmal nocturnal coughing. Initially, the expectoration was scant and mucoid, later becoming greenish, with progressive worsening. He denied fever, dyspnea, chest pain, or systemic symptoms. His medical history included dyslipidemia, well-controlled with a combination of ezetimibe and simvastatin (Vytorin). He had undergone a rhinoseptoplasty and Nissen fundoplication six years prior. He reported no known allergies, smoking history, diabetes, or thyroid dysfunction.

On physical examination, the patient appeared well, with a weight of 74.5 kg and a height of 1.63 m. He was well-hydrated, without signs of cyanosis. Oropharyngeal inspection revealed eutrophic tonsils and mild retropharyngeal hyperemia. There was no jugular venous distension or cervical lymphadenopathy. Lung auscultation was clear bilaterally with normal ventilation. Cardiovascular examination revealed a regular cardiac rhythm without murmurs or additional sounds. The abdomen was soft, non-distended, and non-tender, with mild epigastric discomfort but no signs of gastroesophageal reflux or peritoneal irritation.

Thoracic computed tomography revealed well-expanded lung fields without evidence of consolidation (Figure 1A). However, a well-defined, encapsulated nodular opacity was observed in the right anterosuperior mediastinum (Figure 1B). Notably, the tumor was in close proximity to adjacent vascular structures, but there was no evidence of invasion.

Laboratory tests showed carcinoembryonic antigen (CEA) at 0.95 ng/mL and CYFRA 21.1 at 1.36 ng/mL, both within normal limits. Given the radiologic findings, the patient underwent a right posterolateral thoracotomy. Intraoperatively, a solitary mediastinal mass without significant adhesions was identified and completely resected.

The resected specimen submitted for histopathological evaluation consisted of an irregularly shaped, moderately firm mass with a smooth, grayish-yellow external surface. The mass measured 6 × 5 × 3.5 cm and weighed 58 g. On sectioning, the cut surface revealed a multicystic architecture with white to brown fibrous septations, finely granular internal surfaces, and hemorrhagic contents. The peripheral capsule appeared grossly intact (Figure 2).

Microscopic examination revealed an epithelial neoplasm of neuroendocrine origin. The tumor was composed of cellular nodules exhibiting an organoid and trabecular growth pattern, infiltrating a fibrous connective stroma that demarcated hemorrhagic cavities (Figure 3A). Foci of comedo-type necrosis and irregular zonal necrosis were also identified. The neoplastic nests were bordered by thin collagenous bands and demonstrated prominent apoptotic activity, imparting a characteristic “starry-sky” appearance (Figure 3B). The tumor cells had round to oval nuclei with finely dispersed chromatin and smooth nuclear contours (Figure 3C). The cytoplasm was scant and eosinophilic. Mitoses were observed at a rate of 15 mitoses per mm^2^. Importantly, there was no evidence of invasion into the surrounding fibroadipose tissue, and the tumor was clear of all surgical margins. Immunohistochemical analysis showed coexpression of epithelial markers AE1/AE3 (Figure 3D) and neuroendocrine markers including synaptophysin (Figure 3E), chromogranin A (Figure 3F), INSM1, and CD56 (Figure 3G). Ki-67 immunostaining revealed a proliferation index of 15% (Figure 3H). These findings supported the diagnosis of atypical carcinoid of the thymus. According to the World Health Organization (WHO) classification criteria for NETs [9], which consider both mitotic count and Ki-67 proliferation index, the patient’s tumor was classified as grade 2. Additionally, since the tumor was confined to the thymus and enclosed within an intact capsule, the patient was staged as having Stage I disease [10].

One week after surgery, the patient was clinically stable and in good general condition. He reported only occasional mild pain at the surgical site, with no cough, dyspnea, or fever. Follow-up chest imaging demonstrated well-expanded lung fields with no evidence of pulmonary consolidation, pleural effusion, or residual neoplastic lesions. The aorta appeared prominent and opacified, without evidence of dissection or aneurysm. Cardiac silhouette was within normal limits, with no signs of cardiomegaly. The patient is currently receiving adjuvant chemotherapy and has undergone radiation therapy with a conventional fractionation schedule.

## 3. Discussion

Atypical carcinoid of the thymus is a rare subtype of thymic NET [1]. A large proportion of affected patients are asymptomatic or have nonspecific symptoms [11], as in the case presented here. The most commonly reported symptoms include cough, chest pain, and shortness of breath, which are typically associated with space-occupying lesions in the anterior mediastinum [7]. Although precise epidemiological data are lacking, atypical carcinoid of the thymus tumors are believed to occur more frequently in men than in women [11,12,13], with the average age at diagnosis typically ranging between 40 and 60 years [14,15].

The exact factors that trigger atypical carcinoid of the thymus remain unknown due to the rarity of reported cases, which makes it challenging to establish definitive causes. However, current evidence suggests that the disease is multifactorial, involving a combination of genetic, environmental, and syndromic influences [16]. Among the most frequent mutations are those affecting the menin gene (MEN1), similar to the typical carcinoid of the thymus [17]. MEN1 encodes a transcriptional regulator that is a key component of the MLL/SET1 histone methyltransferase (HMT) complex. This complex specifically catalyzes the methylation of lysine 4 on histone H3 (H3K4), a modification critical for regulating gene expression [18].

The presence of MEN1 gene mutations in atypical carcinoid of the thymus establishes a clear association with MEN1 syndrome, a hereditary neuroendocrine disorder [19]. Similarly, atypical carcinoid of the thymus retain the capacity for hormone production, which can lead to the emergence of paraneoplastic neuroendocrine syndromes. Among the most notable are Cushing’s syndrome, resulting from ectopic adrenocorticotropic hormone (ACTH) secretion [20]. These complications are particularly associated with advanced stages of the disease [13].

Atypical carcinoid of the thymus tumors generally have a poorer prognosis, making early detection essential for effective treatment [21]. However, timely diagnosis is often challenging due to the absence of specific clinical symptoms. As a result, these tumors are frequently not identified until they have progressed to Stage III.

Contrast-enhanced computed tomography is the first tool that can suggest suspicion of this disease [7]. Characteristic findings may include a large, irregular or lobulated mass located in the mediastinum, often adjacent to major blood vessels or involving the pericardium. On non-contrast scans, the tumor may display either uniform or heterogeneous density, while contrast-enhanced images typically reveal mild to moderate heterogeneous enhancement. Additional features that may support the diagnosis include the presence of immature blood vessels and areas of liquefaction or necrosis within the mass [22].

The diagnosis of atypical carcinoid of the thymus primarily depends on pathological examination [21]. Macroscopically, early-stage tumors often appear smooth and well-circumscribed, whereas advanced tumors tend to have an irregular shape and may exhibit calcification. Microscopically, typical thymic carcinoid tumors are well-differentiated, showing minimal necrosis and a low mitotic rate. In contrast, atypical carcinoid tumors are characterized by increased mitotic activity and may show focal necrosis. In the case we present, necrosis was observed, but there was no large increase in mitotic activity, which may be attributed to the tumor being at an early stage of the disease [15]. Immunohistochemistry plays a key role in confirming the diagnosis and distinguishing them from other primary mediastinal masses. Differential diagnosis includes lymphomas, germ cell tumors, parathyroid adenoma or carcinoma, pericardial cysts, thymomas, and metastatic lesions from other primary sites [7]. According to the WHO neuroendocrine tumors of the thymus typically exhibit strong, diffuse immunoreactivity for at least one neuroendocrine marker, such as chromogranin A, synaptophysin, CD56, and neuron-specific enolase (NSE), in more than 50% of tumor cells [9]. The tumor from our patient demonstrated coexpression of epithelial markers AE1/AE3 as well as neuroendocrine markers including synaptophysin, chromogranin A, INSM1, and CD56.

Regarding tumor staging, Stage I tumors are confined to the thymus and are typically well-encapsulated. These tumors are often discovered incidentally or present with mild, nonspecific symptoms such as cough or chest tightness. Histologically, they exhibit 2–10 mitoses per 10 high-power fields (HPF), focal areas of necrosis, and a low-to-intermediate Ki-67 proliferation index (approximately 5–10%). Vascular invasion is rare at this stage. Fortunately, our patient was diagnosed at this early stage of the disease, which is associated with a more favorable prognosis and higher chances of successful treatment. Stage II is characterized by microscopic or macroscopic invasion into the surrounding mediastinal fat or pleura. Clinical symptoms remain subtle, but imaging may reveal irregular tumor margins. Histopathological examination may show early signs of vascular or perineural invasion. In Stage III, the tumor shows gross invasion into neighboring structures, such as the pericardium, great vessels, or lungs. Symptoms become more pronounced and may include chest pain or signs of superior vena cava syndrome. Histologically, there is an increase in mitotic activity, more extensive areas of necrosis, and a Ki-67 index reflecting a higher proliferative rate. Stage IVa involves dissemination to the pleura or pericardium, typically seen as multiple nodules on imaging. Histological features include diffuse infiltration, prominent necrosis, and high mitotic activity, indicating a more aggressive tumor behavior. Stage IVb is defined by distant metastases, most commonly affecting lymph nodes, liver, bone, or lungs. These tumors are highly proliferative, with Ki-67 indices often greater than 20%, and show extensive vascular invasion and overtly aggressive histological characteristics [10]. The most common sites of metastasis include the lymph nodes, bones, lungs, and pleura [13].

The primary treatment for atypical carcinoid of the thymus tumors is complete surgical removal of the tumor. For cases where the tumor is unresectable or has metastasized, the National Comprehensive Cancer Network recommends chemotherapy, with or without radiation therapy [23]. However, the impact of these treatments on long-term patient survival remains uncertain and requires further clinical research [24]. As a preventive measure, our patient received radiotherapy and is currently undergoing chemotherapy to reduce the risk of recurrence. In atypical carcinoid tumors of the thymus, the overall 5-year survival rate ranges from 56% to 77% [2], while the 10-year survival rate drops to approximately 30% [25]. However, in the case we present, the tumor was identified at an early stage and was well encapsulated, suggesting a more favorable clinical prognosis.

Based on our results, we recommend that clinicians maintain a high index of suspicion for atypical carcinoid of the thymus when evaluating mediastinal masses. Although rare, these tumors do occur and should be included in the differential diagnosis of any mediastinal lesion. A timely and accurate diagnosis requires a careful integration of different disciplines. Therefore, close collaboration and effective communication between the treating physician, radiologists, clinical laboratory, surgeons and pathologists are essential to ensure early detection and optimal treatment of the patient.

## 4. Conclusions

This case contributes to the limited but growing body of literature on atypical carcinoid of the thymus and emphasizes the importance of including this rare entity in the differential diagnosis of anterior mediastinal masses. Given the tumor’s nonspecific clinical presentation and potential for aggressive behavior, early recognition is critical. This report underscores the essential role of comprehensive imaging, histopathological evaluation, and immunohistochemical profiling in achieving an accurate diagnosis. Identifying the disease at an early, localized stage—as in this case—can greatly improve the patient’s prognosis and guide more effective, targeted therapeutic strategies. Continued documentation and analysis of such cases are vital to advancing clinical understanding and optimizing management of this rare tumor type.

## Figures and Tables

**Figure 1 medsci-13-00096-f001:**
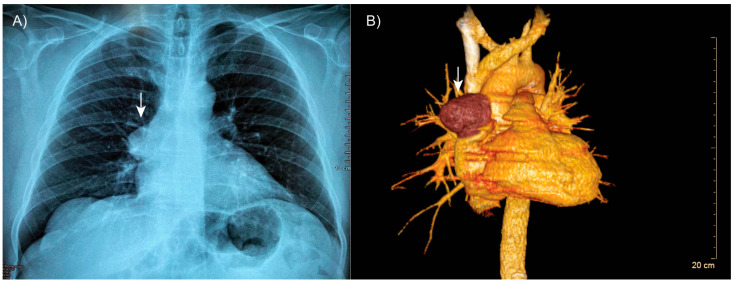
**Patient’s tomography.** (**A**) Thoracic computed tomography showing a well-defined, encapsulated mass in the right anterosuperior mediastinum (arrow). (**B**) Three-dimensional reconstruction demonstrating the lesion (arrow) in close proximity to adjacent vascular structures, with no evidence of invasion.

**Figure 2 medsci-13-00096-f002:**
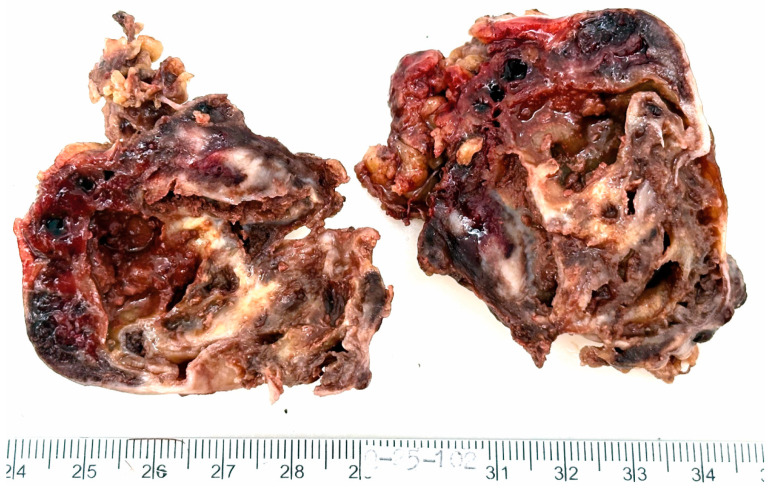
**Macroscopic appearance of the resected tumor.** The mass is multicystic, with white to brown fibrous septations, finely granular inner surfaces and hemorrhagic contents. The external capsule appears intact.

**Figure 3 medsci-13-00096-f003:**
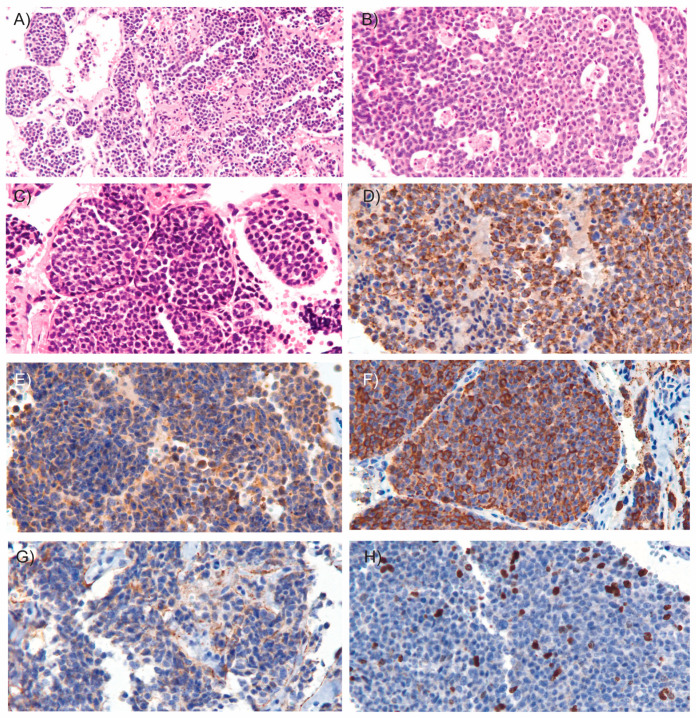
**Microscopic images of the tumor.** Microscopic examination of the resected tumor revealed cellular nodules arranged in an organoid and trabecular growth pattern ((**A**), H&E, 10×), areas of comedo-type necrosis ((**B**), H&E, 15×) and infiltrating a fibrous connective stroma outlining hemorrhagic cavities ((**C**), H&E, 20×). Immunohistochemical staining (20×) demonstrated coexpression of the epithelial marker AE1/AE3 (**D**) and neuroendocrine markers, including synaptophysin (**E**), chromogranin A (**F**), and CD56 (**G**). (**H**) Ki-67 immunostaining shows the tumor’s proliferative activity. In hematoxylin and eosin (H&E) staining, hematoxylin stains acidic structures, such as cell nuclei, blue to purple, while eosin stains basic components, such as cytoplasm, extracellular matrix and connective tissue, various shades of pink to red. In immunostaining, the presence of a target antigen is typically indicated by a brown color resulting from the chromogen DAB (3,3′-diaminobenzidine), with hematoxylin used as a blue nuclear counterstain to provide structural context.

## Data Availability

The original contributions presented in this study are included in the article. Further inquiries can be directed to the corresponding author.

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
