# Peer review of "Atypical Carcinoid of the Thymus: Early Diagnosis in a Case Report"

_medsci, 2025, doi:10.3390/medsci13030096_

Round 1
Reviewer 1 Report
Comments and Suggestions for Authors
This is a case report of an atypical carcinoid of the thymus. I agree with the diagnosis and acknowledge that this tumor type is rare. However, I am not convinced that this case report enhances our understanding of this tumor, as the authors do not seem to offer new insights. Specifically, they emphasize the importance of early detection but do not explain how it can be achieved; in this case, the tumor was found incidentally. For example, the authors could discuss the link between thymic neuroendocrine neoplasms and multiple neuroendocrine syndromes.
Additionally, I would like to comment on some minor issues below.
Page 1, lines 18-19: What are benign thymic lesions? I believe most tumors originating from the thymus are malignant.
Page 1, lines 26-27: “A short summary~” seems unnecessary.
Page 1, line 28: The authors frequently use “atypical thymic carcinoid (tumors)." I suggest using "atypical carcinoid of the thymus" instead.
Page 3, lines 104-105: A microscopic image showing comedonecrosis should be included, as it is crucial for diagnosis.
Page 3, line 108: It should be AE1/AE3, not AE1/AE2.
Figure 3: The current Figure A should be labeled as Figure B, and vice versa, considering the magnification.
Author Response
Dear Reviewer,
We sincerely appreciate the time and effort you have taken to read and review our manuscript, as well as your valuable comments, which have served to improve the clarity and overall quality of our paper. Below you will find my detailed responses to each of your comments, highlighted in blue.
Reviewer 1
This is a case report of an atypical carcinoid of the thymus. I agree with the diagnosis and acknowledge that this tumor type is rare. However, I am not convinced that this case report enhances our understanding of this tumor, as the authors do not seem to offer new insights. Specifically, they emphasize the importance of early detection but do not explain how it can be achieved; in this case, the tumor was found incidentally. For example, the authors could discuss the link between thymic neuroendocrine neoplasms and multiple neuroendocrine syndromes.
As per your suggestion, we have included a discussion on the association between thymic neuroendocrine neoplasms and multiple neuroendocrine syndromes, which now appears in the manuscript as follows:
The presence of MEN1 gene mutations in atypical carcinoid of the thymus establishes a clear association with MEN1 syndrome, a hereditary neuroendocrine disorder [19]. Similarly, atypical carcinoid of the thymus retain the capacity for hormone production, which can lead to the emergence of paraneoplastic neuroendocrine syndromes. Among the most notable are Cushing’s syndrome, resulting from ectopic adrenocorticotropic hormone (ACTH) secretion [20]. These complications are particularly associated with advanced stages of the disease [13].
Additionally, I would like to comment on some minor issues below.
Page 1, lines 18-19: What are benign thymic lesions? I believe most tumors originating from the thymus are malignant.
Thank you for your valuable comment. We have addressed the point you raised, and the revised version now reads as follows:
Despite their rarity, atypical thymic carcinoids are clinically significant due to their aggressive nature and relatively poor prognosis.
Page 1, lines 26-27: “A short summary~” seems unnecessary.
We sincerely apologize for the oversight. It appears that some internal comments intended for communication between co-authors were inadvertently left in the manuscript. These have now been carefully removed.
Page 1, line 28: The authors frequently use “atypical thymic carcinoid (tumors)." I suggest using "atypical carcinoid of the thymus" instead.
We have implemented the modification you noted consistently throughout the manuscript.
Page 3, lines 104-105: A microscopic image showing comedonecrosis should be included, as it is crucial for diagnosis.
An image highlighting comedonecrosis has been added to Figure 3B, as per your valuable recommendation.
Page 3, line 108: It should be AE1/AE3, not AE1/AE2.
It was corrected.
Figure 3: The current Figure A should be labeled as Figure B, and vice versa, considering the magnification.
The figure has been modified according to your suggestion.
Reviewer 2 Report
Comments and Suggestions for Authors
This case report presents the characteristics of a patient with atypical thymic carcinoma. The following issues need to be addressed:
- Please add more detailed information to the title and include "case report."
- Line 26: What does the sentence "A short summary of the article’s main findings" mean? Was it written by AI?
- What are the triggers for atypical thymic carcinoma? What are the common treatment methods?
- How advanced is the patient's atypical thymic carcinoma?
- Figure 1: The color of the arrows in the image is too light; I can hardly see the arrows in image B. Please change the color of the arrows.
- Please indicate what image A represents and what image B represents.
- Figure 3: Please label what the colors represent in the legend.
- The quality of the description of the disease state is good. However, it might be better if Figure 3 included arrows indicating the locations of these pathological features.
- Has the patient's atypical thymic carcinoma reached stage III? Please discuss the characteristics of atypical thymic carcinoma at different stages and compare them with the patient's pathological condition.
- Based on the results of this case report, please provide recommendations for other patients.
- The references were old and can be updated
Author Response
Dear Reviewer,
We sincerely appreciate the time and effort you have taken to read and review our manuscript, as well as your valuable comments, which have served to improve the clarity and overall quality of our paper. Below you will find my detailed responses to each of your comments, highlighted in blue.
Reviewer 2
This case report presents the characteristics of a patient with atypical thymic carcinoma. The following issues need to be addressed:
Please add more detailed information to the title and include "case report."
As per your suggestion, the title has been revised to “Atypical Carcinoid of the Thymus: Early Diagnosis in a Case Report” to provide a more detailed and informative description.
Line 26: What does the sentence "A short summary of the article’s main findings" mean? Was it written by AI?
We sincerely apologize for the oversight. It appears that some internal comments intended for communication between co-authors were inadvertently left in the manuscript. These have now been carefully removed. We confirm that no artificial intelligence tools were used in the preparation of this manuscript.
What are the triggers for atypical thymic carcinoma? What are the common treatment methods?
In response to your insightful questions regarding the triggers of atypical thymic carcinoma, we have added the following comments to the manuscript.
The exact factors that trigger atypical thymic carcinoid remain unknown due to the rarity of reported cases, which makes it challenging to establish definitive causes. However, current evidence suggests that the disease is multifactorial, involving a combination of genetic, environmental, and syndromic influences [16]. Among the most frequent mutations are those of the menin (MEN1) gene, similar to the typical carcinoid of the thymus [17]. MEN1 encodes a transcriptional regulator that is a key component of the MLL/SET1 histone methyltransferase (HMT) complex. This complex specifically catalyzes the meth-ylation of lysine 4 on histone H3 (H3K4), a modification critical for regulating gene ex-pression [18].
Regarding your question about common treatment methods:
The primary treatment for atypical carcinoid of the thymus tumors is complete surgical removal of the tumor. For cases where the tumor is unresectable or has metas-tasized, the National Comprehensive Cancer Network recommends chemotherapy, with or without radiation therapy [21].
How advanced is the patient's atypical thymic carcinoma?
Based on the WHO classification criteria for NETs, which incorporate mitotic count and Ki-67 index, the patient’s tumor was classified as grade 2. This was added to the manuscript:
These findings supported the diagnosis of atypical carcinoid of the thymus. According to the WHO classification criteria for NETs [9], which consider both mitotic count and Ki-67 proliferation index, the patient’s tumor was classified as grade 2. Addi-tionally, since the tumor was confined to the thymus and enclosed within an intact cap-sule, the patient was staged as having Stage I disease [10].
Figure 1: The color of the arrows in the image is too light; I can hardly see the arrows in image B. Please change the color of the arrows.
Please indicate what image A represents and what image B represents.
The figure has been modified by increasing the thickness of the arrow to improve visibility, and the figure legend has been updated according to your suggestion.
Figure 3: Please label what the colors represent in the legend.
The quality of the description of the disease state is good. However, it might be better if Figure 3 included arrows indicating the locations of these pathological features.
As per your kind recommendation, a description of the meaning of the colors in each stain has been added to the figure legend.
Has the patient's atypical thymic carcinoma reached stage III? Please discuss the characteristics of atypical thymic carcinoma at different stages and compare them with the patient's pathological condition.
In response to your valuable comment, we have incorporated the following paragraph into the manuscript.
Regarding tumor staging, Stage I tumors are confined to the thymus and are typically well-encapsulated. These tumors are often discovered incidentally or present with mild, nonspecific symptoms such as cough or chest tightness. Histologically, they exhibit 2–10 mitoses per 10 high-power fields (HPF), focal areas of necrosis, and a low-to-intermediate Ki-67 proliferation index (approximately 5–10%). Vascular invasion is rare at this stage. Fortunately, our patient was diagnosed at this early stage of the disease, which is associated with a more favorable prognosis and higher chances of successful treatment. Stage II is characterized by microscopic or macroscopic invasion into the surrounding mediasti-nal fat or pleura. Clinical symptoms remain subtle, but imaging may reveal irregular tu-mor margins. Histopathological examination may show early signs of vascular or peri-neural invasion. In Stage III, the tumor shows gross invasion into neighboring structures, such as the pericardium, great vessels, or lungs. Symptoms become more pronounced and may include chest pain or signs of superior vena cava syndrome. Histologically, there is an increase in mitotic activity, more extensive areas of necrosis, and a Ki-67 index often exceeding 10%, reflecting a higher proliferative rate. Stage IVa involves dissemination to the pleura or pericardium, typically seen as multiple nodules on imaging. Histological features include diffuse infiltration, prominent necrosis, and high mitotic activity, indi-cating a more aggressive tumor behavior. Stage IVb is defined by distant metastases, most commonly affecting lymph nodes, liver, bone, or lungs. These tumors are highly prolifera-tive, with Ki-67 indices often greater than 20%, and show extensive vascular invasion and overtly aggressive histological characteristics [10].
Based on the results of this case report, please provide recommendations for other patients.
In accordance with your suggestion, we have included some recommendations:
Based on our results, we recommend that clinicians maintain a high index of suspicion for atypical carcinoid of the thymus when evaluating mediastinal masses. Although rare, these tumors do occur and should be included in the differential diagnosis of any mediastinal lesion. A timely and accurate diagnosis requires a careful integration of different disciplines. Therefore, close collaboration and effective communication between the treating physician, radiologists, clinical laboratory, surgeons and pathologists are essential to ensure early detection and optimal treatment of the patient.
The references were old and can be updated
Due to the limited number of available cases, some of the references cited are somewhat older. However, we have retained them given their relevance and contribution to the field. We would like to highlight that 50% of the references are from the past five years, reflecting the incorporation of recent and up-to-date literature.
Reviewer 3 Report
Comments and Suggestions for Authors
Dear Authors,
I think that the Manuscript is very interesting and well written.
Round 2
Reviewer 2 Report
Comments and Suggestions for Authors
The manuscript is approved.